# Coupled Preparation of Ferronickel and Cementitious Material from Laterite Nickel Ores

**DOI:** 10.3390/ma13214992

**Published:** 2020-11-05

**Authors:** Ruimeng Shi, Xiaoming Li, Yaru Cui, Junxue Zhao, Chong Zou, Guibao Qiu

**Affiliations:** 1School of Metallurgical Engineering, Xi’an University of Architecture and Technology, Xi’an 710055, China; xmli@xauat.edu.cn (X.L.); yaroo@126.com (Y.C.); Zhaojunxue1962@126.com (J.Z.); zouchong985@163.com (C.Z.); 2College of Materials Science and Engineering, Chongqing University, Chongqing 400044, China

**Keywords:** laterite nickel ore, ferronickel, C_3_S (tricalcium silicate), C_3_A (tricalcium aluminate), cementitious material, process coupling

## Abstract

Nickel slags can be produced through ferronickel preparation by the pyrometallurgical processing of laterite nickel ores; however, such techniques are underutilized at present, and serious environmental problems arise from the stockpiling of such nickel ores. In this study, a modification to the process of ferronickel preparation by the direct reduction of carbon bases in laterite nickel ores is proposed. The gangue from the ore is used as a raw material to prepare a cementitious material, with the main components of tricalcium silicate and tricalcium aluminate. By using FactSage software, thermodynamic calculations are performed to analyze the reduction of nickel and iron and the effect of reduction on the formation of tricalcium silicate and tricalcium aluminate. The feasibility of a coupled process to prepare ferronickel and cementitious materials by the direct reduction of laterite nickel ore and gangue calcination, respectively, is discussed under varying thermodynamic conditions. Different warming strategies are applied to experimentally verify the coupled reactions. The coupled preparation of ferronickel and cementitious materials with calcium silicate and calcium aluminate as the main phases in the same experimental process is realized.

## 1. Introduction

Nickel (Ni) is a transition element that exhibits a mixture of ferrous and nonferrous metal properties, and is used in many industries, for example for making stainless steel, superalloy, non-ferrous alloy, superalloy, coin, and batteries [1,2,3,4,5]. World nickel reserves are abundant [6]. Identified land-based resources averaging 1% nickel or greater contain at least 130 million tons of nickel, with about 60% in laterites and 40% in sulfide deposits. In 2019, nickel resources are divided into two types of ores: sulfide and laterite, of which sulfide ore accounts for about 30% and laterite nickel ore accounts for the rest [7,8,9]. In China, about 70% of nickel reserves are concentrated in Gansu Province, next to Canada’s Sadbury nickel, Asia rank the first and world rank the second, followed by Xinjiang, Yunnan, Jilin, and other places [10,11,12]. Before 2000, because of available upgrading methods, availability of high-grade ore, and low energy consumption in the extraction process, nickel was mainly produced from sulfide (60%), while laterite needed comprehensive treatment to extract nickel metal, resulting in a higher processing cost of laterite than nickel sulfide [13,14,15]. In recent years, with the rapid development of our national economy, the consumption of nickel is increasing day by day. At the same time, the total amount and grade of nickel sulfide ore resources shows a downward trend, and laterite is becoming increasingly important in the future. Therefore, pursuers begin to improve the nickel recovery technology of laterite nickel ore [16,17]. Laterite nickel ore has low exploration cost, mature technology, and broad development prospects. Therefore, nickel oxide mining is listed as a key project in China’s “medium and long term scientific development and regulation of non-ferrous metal industry (2006–2020)” (draft for comments) [18,19].

The nickel content of laterite ores is very low, and special treatments are needed to improve the nickel content of these ores, such as physical separation techniques, but the complexity of laterite nickels makes it difficult to achieve any significant breakthroughs [20,21,22,23,24,25]. Thermal modification of laterite ore by carbonization, dehydration, and reduction technologies has been shown to significantly improve nickel extraction in the acid leaching stage [26,27,28,29,30,31,32,33]. Pyrometallurgy is the main process route for nickel extraction from saprolite ore, and there are two commonly used laterite nickel smelting methods [34,35]. One is the rotary kiln dry-electric furnace reductive smelting. The laterite nickel ore is placed in the rotary kiln for drying intervention reduction at a temperature of 650–800 °C, and then placed in the reducing furnace at 1550–1600 °C for smelting and sorting to obtain coarse nickel iron. Nickel–iron alloys containing more than 25% nickel can be produced after further refining [36]. The second method is rotary kiln direct reduction. After drying, crushing, and sieving, the raw ore is mixed with lime and pulverized coal in proportion to the pellet. Drying and high temperature reduction produce spongy nickel–iron alloys and slag mixtures. After cooling, grinding, screening, and magnetic separation, the coarse nickel–iron particles were obtained. However, there are some problems in laterite nickel mining technology. In the process of rotary kiln drying-furnace reduction smelting, the melting temperature is high, the power consumption is large, the overall energy consumption is large, the slag production is large, the direct use failure is serious. Due to the direct reduction of rotary kiln, tailings yield is large and slag cannot be directly used, which increases the cost of subsequent treatment of nickel slag [37,38,39,40,41,42,43,44,45]. However, in the process of fire smelting of nickel–iron from laterite, the gangue components in the raw ore are comprehensively treated and applied to high value-added materials, so as to realize the green metallurgical utilization of laterite nickel, which has the characteristics of low cost, high added value, and short flowing time. There are great similarities between technology and equipment in traditional rotary kiln production of direct reducing iron and cement clinker production. First of all, the main equipment is the same rotary kiln, and the raw materials for the production of cement clinker are similar to the gangue composition in iron ore. Secondly, their main production processes are the same. All raw materials must be processed by the process of abrasive mixing—roasting—abrasive, and the roasting temperature is relatively close. The difference is that in the production process of direct reduced iron, it is necessary to separate the direct reduced iron from the gangue components after grinding by means of magnetic separation, while the purpose of cement clinker abrasive is to control the size of cementing materials. In the traditional direct reduced iron production process, gangue formation consumes a lot of energy in each process, but it cannot form high value-added products in the end. Instead, it is treated as solid waste.

Due to the correlation between the two production processes, we proposed the idea of coupling the process of preparing direct reduced iron with the process of preparing cementitious materials. The purpose of the experiment is to design a reaction coupling process. In the process of preparing direct reduced iron or nickel–iron alloy, it can also make the gangue component in the ore react with the CaO flux to produce the cementitious material. In a material flow and energy flow process, two different valuable products of nickel–iron alloy and cementitious material can be obtained without producing solid waste. Compared with the traditional process, the new process has the characteristics of high energy efficiency, high resource efficiency, and no solid waste. In this study, while preparing nickel iron from carbon base of laterite nickel ore by direct reduction, the gelling materials with silicate tricalcium silicate (C_3_S) and aluminate tricalcium (C_3_A) as main components were prepared from coal gangue. The coupling of these two reaction processes was investigated experimentally.

## 2. Material Analysis and Burdening

In the experiment, X-ray diffraction (XRD) analysis and fluorescence spectroscopy chemical component analysis were performed on the raw laterite nickel ore (Jinchuan Group Ltd., South Halmahera, Indonesia). The X-ray diffractometer used for detection is the Empyrean type with the scanning rate of 8°/min from PANalytical B.V (Almelo, Netherlands). The XRD analysis result from the laterite nickel ore is shown in Figure 1, and the chemical components determined through fluorescence spectroscopy are listed in Table 1.

The figure shows that the main phases of the laterite nickel ore are FeO(OH) (Goethite), Al(OH)_3_ (Gibbsite), and Fe_2_O_3_. The XRD peaks for the SiO_2_ phase are comparatively lower, and those for other minerals with calcium, magnesium, and nickel are not obvious. The laterite nickel ore thus has FeO(OH), Al(OH)_3_, and Fe_2_O_3_ as the main components.

According to XRD analysis and fluorescence spectroscopy, in the laterite nickel ore, FeO(OH) and Fe_2_O_3_ are the main phases. Among them, the content of Fe_2_O_3_ reaches 46.26%, the content of SiO_2_ is 3.30%, and the content of Al(OH)_3_ is equivalent to Al_2_O_3_, which accounts for 8.77% of the total mass, while the content of Ca, Mg, Ni, and other elements is less. Based on these characteristics of phase composition, through thermodynamic analysis and calculation, the process coupling for the reduction of nickel and iron and the formation of C_3_S, C_3_A, and other cementitious materials via the reaction of CaO with SiO_2_ or Al_2_O_3_ is calculated. Other elements with lower contents are not considered in the thermodynamic analysis.

Based on the above analysis, the mix ratio of reactants in the raw material is calculated first. The principle for burdening mainly includes two aspects: (1) The addition of reducing agent must ensure the complete reduction of nickel and iron; (2) the added amount of CaO must be sufficient to fully react with SiO_2_ and Al_2_O_3_ to produce C_3_S and C_3_A. According to these calculations, the ratio of laterite nickel ore and raw materials for reduction roasting is shown in Table 2. All the raw materials are ground to particles of less than 200 mesh in size. Burdening was prepared in conformity with the calculated ratio and reaction samples were made by fully blending. The thermodynamic analysis and calculation were performed based on this sample’s ratio of raw materials.

## 3. Thermodynamic Analysis of Coupling Reaction

In the reduction roasting process, chemical reactions including the decomposition of FeO(OH) (Goethite) and Al(OH)_3_ (Gibbsite), reduction of nickel and iron, reaction of iron oxides with CaO to form calcium ferrite, reaction of CaO with Al_2_O_3_ to form CaAl_2_O_4_ (CA) and Ca_3_Al_2_O_6_ (C_3_A), reaction of CaO with SiO_2_ to create Ca_3_SiO_5_ (C_3_S, alite) and Ca_2_SiO_4_ (C_2_S, Belite), and reaction of CaO with C_2_S to create C_3_S may occur. FeO(OH) and Al(OH)_3_ can decompose into Fe_2_O_3_ and Al_2_O_3_ after heating, respectively; thus, Fe_2_O_3_ and Al_2_O_3_ are used to replace the original phases in thermodynamic analysis and calculation, following the relevant chemical reaction Equations (1)–(9). Using the Reaction model in FactSage software (FactSage 6.3, Thermfact/CRCT, Montreal, Canada and GTT-Technologies, Aachen, Germany), the possible standard Gibbs free energy produced by the reactions among the main phases during reduction roasting is calculated at intervals of 10 °C between 0 °C and 2000 °C, and the relationship of temperature and standard Gibbs free energy Δ*G* for each equation is drawn via Origin (Origin 2019, OriginLab, Northampton, MA, USA) as shown in Figure 2 [46].
NiO + C→Ni + CO(1)
Fe_2_O_3_ + 3C→2Fe + 3CO(2)
Fe_2_O_3_ + 2CaO→Ca_2_Fe_2_O_5_(3)
CaO + Al_2_O_3_→CaAl_2_O_4_(4)
2CaO + CaAl_2_O_4_→Ca_3_Al_2_O_6_(5)
3CaO + Al_2_O_3_→Ca_3_Al_2_O_6_(6)
2CaO + SiO_2_→Ca_2_SiO_4_(7)
3CaO + SiO_2_→Ca_3_SiO_5_(8)
CaO + Ca_2_SiO_4_→Ca_3_SiO_5_(9)

As shown in Figure 2a, in the temperature range 500–1450 °C, Equations (1), (3), and (8) proceed spontaneously to realize the reduction of nickel and the production of CA, C_3_A, C_2_S, C_3_S, and Ca_2_Fe_2_O_5_. In Equation (2), the temperature at the beginning of iron reduction is 653 °C. At temperatures below 780 °C, the reaction of Equation (3) consumes CaO and Fe_2_O_3_ and then produces Ca_2_Fe_2_O_5_; because the reaction proceeds more easily than iron reduction in Equation (2), this temperature regime is unfavorable for iron reduction. At temperatures above 780 °C, the reaction of Equation (2) proceeds before that of Equation (3); Fe_2_O_3_ is reduced to iron and the temperature for Fe_2_O_3_ reduction should be higher than 780 °C, under which nickel reduction can also occur.

Through the comparison of the ΔG–T curves of Equations (4)–(6) in Figure 2a, when the temperature is above 500 °C, CaO and Al_2_O_3_ react to produce C_3_A more easily. The product of CA also reacts further with CaO to produce C_3_A; therefore, at high temperatures, the reaction of CaO with Al_2_O_3_ finally produces the C_3_A phase.

Through the comparison of the ΔG–T curves of Equations (7) and (8) in Figure 2a, at 1300 °C, the reaction given by Equation (7) proceeds more easily compared to that given by Equation (8) to produce C_2_S. At temperatures above 1300 °C, C_3_S is more easily produced. From Figure 2b, only at temperatures between 1300 °C and 1800 °C does the reaction of Equation (9) occur spontaneously; then, C_2_Se produced early in the reaction reacts with calcium oxide to generate C_3_S. Combining Figure 2a,b, to acquire a cementitious material with C_3_A and C_3_S as the main phases, the temperature must be between 1300 °C and 1800 °C.

To discuss the reactant phase compositions under high temperatures when iron is not fully reduced, the phase model in FactSage software is used to determine the phase diagram for the quaternary oxide system of CaO–SiO_2_–Al_2_O_3_–Fe_2_O_3_ at 1350 ℃. Supposing that unreduced iron comprises 10% of the total mass of this quaternary oxide system, the resulting phase diagram is as shown in Figure 3.

The red line in Figure 3 indicates the raw material ratio of Al_2_O_3_/SiO_2_ = 2.66, equal to that in the laterite nickel ore. The phase composition of the quaternary oxide system varies along the red line as the CaO content changes. For a continuous increase in CaO content, the phase composition of the quaternary oxide system also changes correspondingly. For lower CaO contents (area 4 in the figure), the phase composition includes CA, Ca_2_SiO_4_, Ca_2_Al_2_SiO_7_, and Ca_2_Fe_2_O_5_. As the CaO content increases (area 3), CaO reacts with Ca_3_Al_2_SiO_7_ to produce C_3_A and C_2_S, and the Ca_3_Al_2_SiO_7_ phase disappears. With further increases in CaO content (area 2), sufficient CaO reacts with CA to form C_3_A and consume CA; afterward, CaO reacts with C_2_S to generate C_3_S. When the CaO content reaches area 1, C_2_S is fully consumed, for it reacts completely with CaO to produce C_3_S. Additional CaO exists in the free state.

In addition, Fe_2_O_3_ reacts with CaO at 1350 °C to produce Ca_2_Fe_2_O_5_, which coexists with CA, C_3_A, C_2_S, and C_3_S (areas 1–4) as impurities in cementitious materials. Thus, Fe_2_O_3_ should be reduced to iron to avoid yielding Ca_2_Fe_2_O_5_.

Figure 3 indicates that a sufficiently high CaO content (areas 1–3) can ensure the coexistence of silicate and aluminate systems independently under high temperatures. To further explore the interaction relationship of reduced iron with the silicate and aluminate systems, FactSage is used to determine the two ternary diagrams of Fe–CaO–Al_2_O_3_ and Fe–CaO–SiO_2_. Iron is set to comprise 50% of the whole, as shown in Figure 4 and Figure 5.

Figure 4 demonstrates that C_3_A is somewhat unstable unless it is produced within the temperature range 237–1540 °C. Stable C_3_A is produced by the reaction of aluminum oxide with CaO (areas 3–5). At temperatures below 237 °C (area 6), C_3_A is decomposed into CA. At temperatures higher than 1540 °C (area 1), C_3_A reacts with the slag liquid phase and decomposes. Reduced iron is not obviously influential on the CaO–Al_2_O_3_ system, indicating that they can coexist. For temperatures between 237 °C and 1540 °C and lower CaO contents (CaO/Al_2_O_3_ < 0.55), no C_3_A forms in the system. The reaction of CaO with Al_2_O_3_ under these conditions produces CaAl_4_O_7_ and CaAl_2_O_4_, which have lower calcium contents (area 7); as the CaO content increases to 0.55 < CaO/Al_2_O_3_ < 1.65 (area 7), the low-calcium-content CaAl_4_O_7_ phase disappears and the more calcium-rich Ca_3_Al_2_O_6_ phase is produced. CaAl_2_O_4_ ceases forming for CaO/Al_2_O_3_ > 1.65, at which point the main phases are C_3_A, free-state CaO, and coexistent iron (area 5), meaning that CaO/Al_2_O_3_ = 0.55 and CaO/Al_2_O_3_ = 1.65 are the composition points of CaAl_2_O_4_ and Ca_3_AlO_6_, respectively. In conclusion, when cementitious materials with the main phase of C_3_A are prepared, the effective mass of CaO combined with Al_2_O_3_ should meet the proportion of CaO/Al_2_O_3_ = 1.65 in the blending process, and the roasting temperature should be between 237 °C and 1540 °C.

Figure 5 indicates that C_3_S is also unstable outside of the temperature range 1300 °C to 1800 °C. Stable C_3_S is produced by the reaction of aluminum oxide with C_2_S (areas 2–4) within this range. When the reaction temperature is lower than 1300 °C or higher than 1800 °C (areas 1 and 5), C_3_S is decomposed to C_2_S and free-state CaO, consistent with the thermodynamic calculations shown in Figure 2b. Meanwhile, the effect of reduced iron on the CaO–SiO_2_ system is not significant, indicating that these three phases can coexist. At temperatures 1300 °C–1800 °C and lower CaO contents (CaO/SiO_2_ < 1.40), neither C_3_S nor C_2_S is found in the system. The reaction of CaO with SiO_2_ mainly produces CaSiO_3_ and Ca_3_Si_2_O_7_. When the ratio of CaO/SiO_2_ is between 1.40 and 1.86, C_2_S exists in the system (areas 7 and 8); when the ratio range is 1.86 to 2.79, C_2_S and C_3_S can coexist in the system (areas 3 and 4); when the ratio range exceeds 2.79, C_3_S, free-state CaO, and coexistent Fe (area 2) are the main phases, meaning that CaO/SiO_2_ = 1.40 and CaO/SiO_2_ = 1.86 are the composition points of Ca_3_Si_2_O_7_ and C_2_S, respectively. In conclusion, when the cementitious materials with the main phase of C_3_S are prepared, the effective mass of CaO combined with SiO_2_ should meet the proportion CaO/SiO_2_ = 2.79 in the blending process, and the roasting temperature should be between 1300 °C and 1800 °C.

With powdered carbon as the reducing agent, laterite nickel ore as the raw material, and CaO as the additive, the theoretical analysis results of the changes in the phase composition of the reactants during heating are calculated under a protective CO atmosphere, by the equipment module in the FactSage software. As shown in Table 2, the reactant proportion was the same. Figure 6 shows the calculation result of the theoretical reaction product, which analyzes the phase change of the system from 500 °C to 1500 °C in a temperature interval of 50 °C.

It can be seen from Figure 6 that in the reaction in the low temperature zone (<569 °C), the phases formed are mainly carbon, pyroxene, CaCO_3_, Fe_2_O_3_, nickel, and partly reduced iron. When the temperature exceeds 569 °C, a liquid iron–nickel phase is formed with a high content of nickel; the ratio of this liquid alloy phase increases with increasing of temperature. At temperatures above 653 °C, most of the Fe_2_O_3_ and carbon reactants are consumed to form iron. For temperatures above 787 °C, CaCO_3_ is decomposed into CaO, part of which reacts with SiO_2_ and Al_2_O_3_ to generate C_2_S and some Ca_3_Al_2_O_6_, while the pyroxene is consumed. When the reaction temperature is between 1093 °C and 1300 °C, there is no obvious change in each phase, and the main components of liquid iron–nickel alloy, iron, CaO, Ca_3_Al_2_O_6_, and C_2_S. However, at temperatures above 1300 °C, C_3_S is formed by the reaction of CaO and C_2_S. Under high-temperature reaction, the final products are liquid iron–nickel alloy, Ca_3_Al_2_O_6_, and C_3_S.

According to the above-mentioned thermodynamic analysis, it is feasible to prepare a cementitious material of C_3_A and C_3_S by the reduction roasting of laterite nickel ore after the appropriate proportioning. However, in order to ensure the sufficient reduction of nickel and iron as well as the effective production of C_3_A and C_3_S, the roasting temperature must be controlled by stages.

The reduction temperature of Fe_2_O_3_ should be controlled at 780–1220 °C since Fe_2_O_3_ reacts with CaO to form Ca_2_Fe_2_O_5_ which affects iron reduction. Thus, in order to ensure complete iron reduction and prevent the formation of low-melting-point liquid fayalite (which melts at 1220 °C) by Fe_2_O_3_ and SiO_2_, this temperature should be maintained for a certain period of time. Under these conditions, C_2_S and a small amount of Ca_3_Al_2_O_6_ can be produced. After the sufficient reduction of iron, the temperature is increased again and then maintained at 1300–1540 °C (the temperature range for forming stable existent C_3_S is 1300–1800 °C, that for the decomposition of pyroxene is 1093 °C, and that for stable existent C_3_A is 237–1540 °C). In this regime, the formation of C_3_S and the reaction of pyroxene after decomposition with CaO to form Ca_3_Al_2_O_6_ are accomplished. Finally, the samples are cooled by fast cooling; tricalcium silicate is metastable at normal temperatures with this treatment.

## 4. Experiments and Results

To verify the accuracy of the above thermodynamic analyses and calculations, coupled experimental studies are conducted under different reaction conditions. Three different control processes are set using three 8 g samples placed in separate Al_2_O_3_ crucibles. Each crucible is placed in airtight tube-type resistance furnace with the heating rate set to 10°C/min. The controlling processes for the three samples are as follows:No atmospheric protection; the temperature is increased from indoor ambient temperature to 1450 °C and maintained for 1 h before cooling to indoor temperature to obtain product A.At the flow rate of 60 mL/h under the atmospheric protection of CO, the temperature continues to rise from room temperature to 1000 °C, maintained for 2 h, and then increased to 1450 °C and maintained for 1 h. Product B was obtained after furnace cooling to room temperature.The period of cooling applied in (2) is changed. The atmosphere conditions and control process of the temperature increase to 1450 °C are the same as those used for obtaining sample B. After maintaining the temperature of 1450 °C for 1 h, by using fast cooling, product C was obtained.

Finally, the obtained reaction samples were examined by XRD and scanning electron microscopy-energy-dispersive X-ray spectroscopy (SEM-EDS). The XRD equipment was the same as that used for raw material analysis; the SEM analysis equipment was type VEGA II XMU (TESCAN Company, Brno, Czechia); the EDS analysis equipment was type 7718 (Oxford Instruments, London, UK). The XRD and SEM-EDS analysis results are shown in Figure 7 and Figure 8, respectively.

Iron oxide comprises more than 45% of the total sample mass, and the ferronickel alloy is 40% of the total mass in the theoretical reaction product. Therefore, it can be seen from the XRD pattern that the intensity of the diffraction peaks of iron–nickel alloy and iron oxide is higher than that of other phases.

As can be seen from the XRD analysis results, sample A shows incomplete iron reduction, attributed to the lack of atmosphere protection; iron is present as unreacted Fe_2_O_3_ and Ca_2_Fe_2_O_5_ after reacting with CaO. The consumption of CaO causes the reduction of CaO/Al_2_O_3_ and CaO/SiO_2_. The diffraction peaks of the low calcium phase appear obviously in the XRD pattern of the reaction products, such as CaAl_2_O_4_ and Ca_2_SiO_4_. At the same time, the diffraction peaks of Ca_3_Al_2_O_6_ have a weak diffraction peak, which is consistent with the analysis results in Figure 3. At 1000 °C for 2h, sample B reacts in a reducing atmosphere to ensure that the nickel–iron is completely reduced to form an iron–nickel alloy, and is then held at 1450 °C for 1 h to form C_3_S. However, as a result of furnace cooling, C_3_S is decomposed into C_2_S and CaO phases. Therefore, the final products of sample B are iron, CaO, Ca_2_SiO_4_, CaAl_2_O_4_, and Ca_3_Al_2_O_6_, which are consistent with the analysis results of Figure 4 and Figure 5. Sample C uses the same heating control method as sample B under the protection of CO to achieve the reduction of nickel–iron to form iron–nickel alloys, C_3_S and C_3_A. C_3_S avoids the large-scale decomposition using rapid cooling. The final products are iron, C_3_S, and Ca_3_Al_2_O_6_, and the decomposition products are CaAl_2_O_4_ and Ca_2_SiO_4_, which show obviously stronger XRD peaks than that of Ca_2_SiO_4_ in sample B. Simultaneously, the SEM-EDS analysis and scanning of sample C show that the morphology of the product is composed of many iron grains with different grain diameters and the cementing material in them. The distributions of nickel and iron are the same; those of calcium, aluminum, silicon, and oxygen are the same, and clear boundaries are observed between the ferronickel alloy and the cementitious materials. The analytical results of theoretical products shown in Figure 6 are consistent with the above analytical results.

## 5. Conclusions

For the preparation of high-temperature coexisting cementitious materials such as ferronickel alloy, C_3_S, C_2_S, etc., it is feasible to reduce and roast laterite nickel ore after proportioning.It is necessary to control the coupling reaction temperature in stages. The reduction temperature of Fe_2_O_3_ is controlled at 780–1220 °C, the formation temperature of C_3_S is controlled at 1300–1800 °C, and the stability of C_3_S is controlled at 237–1540 °C. Fast cooling must be applied in order to obtain a metastable C_3_S phase.At the reducing atmosphere protection of CO, the first reaction stage is holding at 1000 °C for 2 h, the second reaction stage is holding at 1450 °C for 1 h, and fast cooling is used. In this process, ferronickel alloy, C_3_S, and Ca_3_Al_2_O_6_ laterite nickel ore are reduced and roasted, and the decomposition products are Ca_2_SiO_4_ and CaAl_2_O_4_ to obtain the final products.The reduction of iron has a significant effect on the formation of C_3_S. When iron is insufficiently reduced, calcium ferrite and other phases are formed in combination with CaO, hindering the formation of C_3_S. The necessary condition for the formation of C_3_S is the sufficient reduction of iron.

## Figures and Tables

**Figure 1 materials-13-04992-f001:**
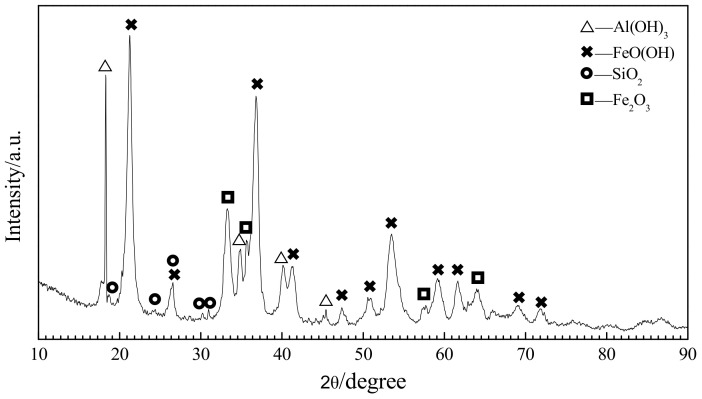
Laterite nickel ore XRD analysis.

**Figure 2 materials-13-04992-f002:**
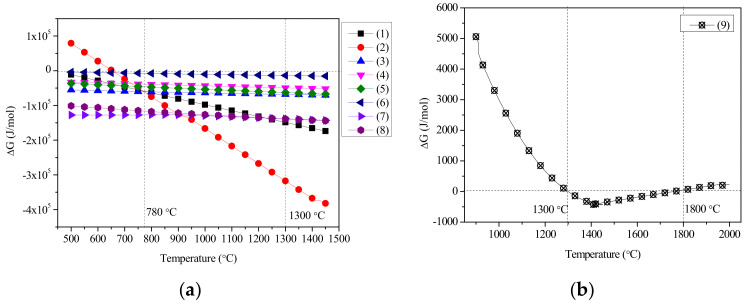
Laterite nickel ore coupling reaction Δ*G*–*T*: (**a**) Equations (1)−(8); (**b**) Equation (9).

**Figure 3 materials-13-04992-f003:**
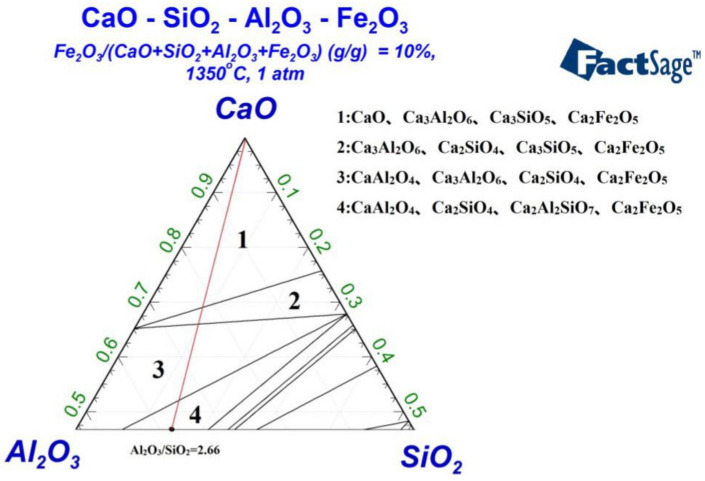
Phase diagram of quaternary oxide system CaO–SiO_2_–Al_2_O_3_–Fe_2_O_3_ at 1350 °C.

**Figure 4 materials-13-04992-f004:**
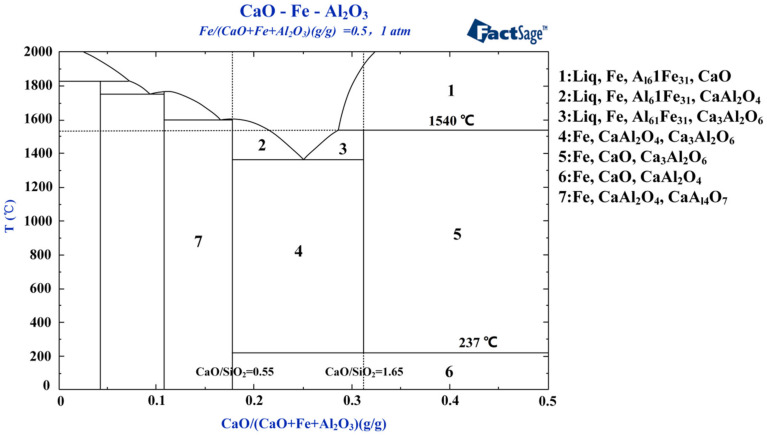
Ternary diagram of Fe–CaO–Al_2_O_3_.

**Figure 5 materials-13-04992-f005:**
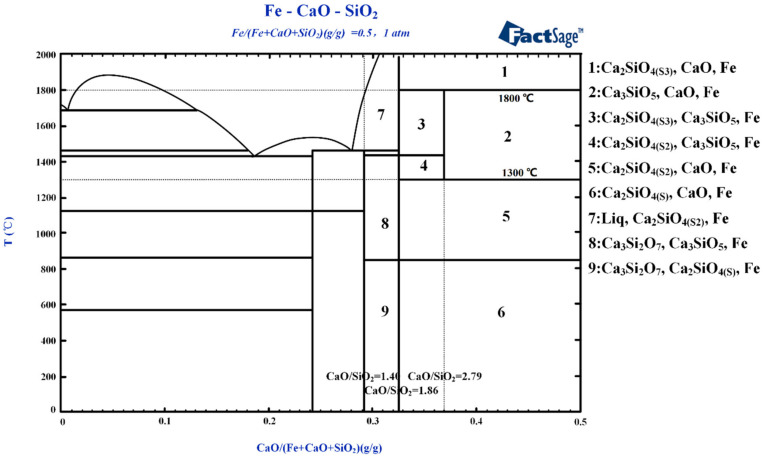
Ternary phase diagram of Fe–CaO–SiO_2_ [46].

**Figure 6 materials-13-04992-f006:**
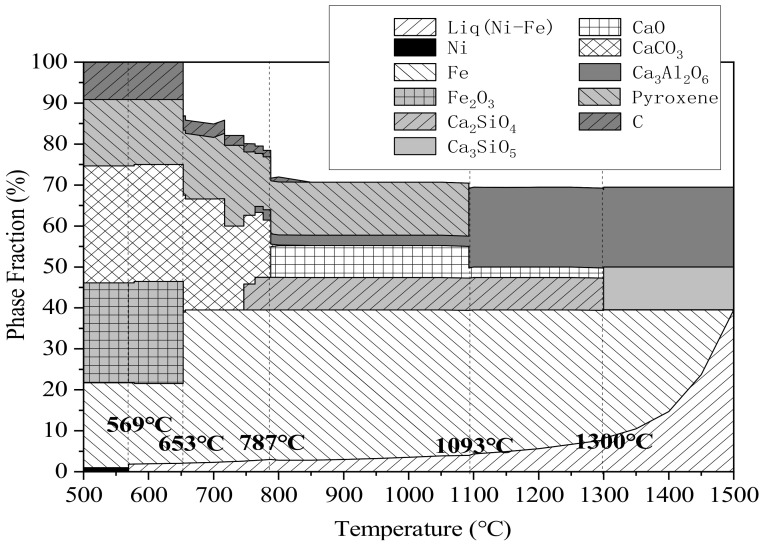
Diagram of variations in laterite nickel ore reduction roasting products.

**Figure 7 materials-13-04992-f007:**
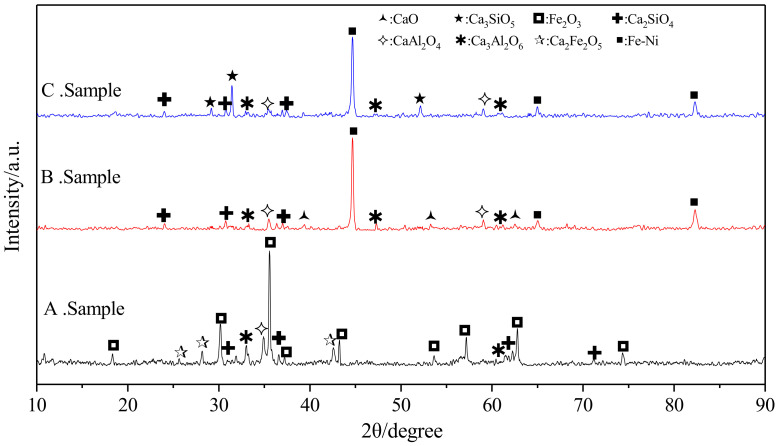
XRD analysis for reduction roasting products of laterite nickel ore.

**Figure 8 materials-13-04992-f008:**
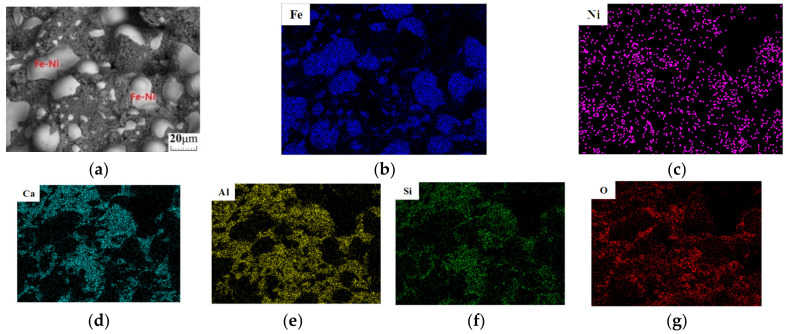
SEM–EDS analysis chart of sample C: (**a**) SEM analysis; (**b**) EDS analysis of Fe; (**c**) EDS analysis of Ni; (**d**) EDS analysis of Ca; (**e**) EDS analysis of Al; (**f**) EDS analysis of Si; (**g**) EDS analysis of O.

**Table 1 materials-13-04992-t001:** Chemical components of laterite nickel ore (%wt.).

Components	NiO	Fe_2_O_3_	FeO	SiO_2_	Al_2_O_3_	CaO	MgO	Water
Content	1.54	65.23	0.44	3.30	8.77	0.96	1.51	17.31

**Table 2 materials-13-04992-t002:** Laterite nickel ore reduction roasting burden sheet (%wt.).

Composition	Laterite Nickel Ore	CaO	C
Ratio	68.42	19.03	12.55

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
