# Peer review of "Coupled Preparation of Ferronickel and Cementitious Material from Laterite Nickel Ores"

_materials, 2020, doi:10.3390/ma13214992_

Round 1
Reviewer 1 Report
The research paper is interesting and offers new knowledge in the field of preparation of cementitious material from laterite nickel ores .
However, it should be corrected and amended, especially the part Experiments and Results before publishing in the journal. I suggest to authors to make following corrections of the manuscript:
1. Introduction - First paragraph - Once you use "percent" and in another place "%", you should unify.
2. Why is this gangue used for cementitious material with the main components of tricalcium silicate (C3S) and tricalcium aluminate (C3A), why
is not the main components C2S or C4AF ? Can you specific more ?
3. Material Analysis and Burdening
- from which area the laterite ore comes ?
- Table 1 is very confusing - the sum of all stated oxides is more than 100%, What exactly does TFe mean? Instead of water, I suggest to use H2O
- line 74-75 - Try to transform the sentence to better understandable form (aluminum is not converted, but bonded in Al2O3, etc.)
- Do you have also particle size distribution of the laterite ore and density ?
- The thermodynamic analysis and the calculations were realised according some previous literature or can you state some equations for the calculation
of the exact amount of CaO that must be sufficient to fully react with SiO2 and Al2O3 to produce C3S and C3A ?
- Table 2 - can you specific "C" (carbon) ?
3. Thermodynamic Analysis of Coupling Reaction
- what about C4AF ? you did not count with ferrite ?
- Corect "Tempreature" to "Temperature" in Fig. 2
- Line 105 - How did you find out the temperature 653 °C?
- Statements in lines 103-126 should be discussed with some literature
- Line 124 - Shouldn´t it be Fe2O3 insted of Fe2O? (quaternary oxide system)
- Line 133 - Shouldn´t it be "Ca2Al2SiO7" insted of "Ca3Al2SiO7" ?
- Figure 4, 5 - "y" axis - (°C) instead of (C), comma separation of individual formulations (1-7), Figure 5 - check lower indexes
4. Experiments and Results:
- Line 263 - "The distributions of nickel and iron are the same." - this is not a true, please check again SEM-EDS.
- Fig. 8 - Can you mark iron shots in SEM photomicrograph ? What do you mean by "cementitious materials" (line 263)? (CSH pahses, portalndite, etc. ?)
- Can you also mark boundaries between the ferronickel alloy and the cementitious materials in the SEM photomicrographs ?
- Doy you have SEM-EDS only for the sample C ?
Line 273 - check lower index
Line 269 - What are you meaning by other cementitious materials ? Are you meaning hydrated minerals (components) of the cement or another materials based on a cement?
Reviewer 2 Report
The manuscript presents an adequate thermodynamic study in terms of phase equilibrium during a laterite ore reduction. However, unfortunately, it cannot be published as the whole approach of producing a complex FeNi-ceramic material is, under my point of view, meaningless, for the following reasons:
1.The thermodynamic study presented has no added value, as it is not presenting anything new, compared to other previous publications on the same issue using also the FactSage software (M. A. Rhamdhani, P. C. Hayes and E. Jak, “Nickel laterite Part 2 – thermodynamic analysis of phase transformations occurring during reduction roasting”, Mineral Processing and Extractive Metallurgy (Trans. Inst. Min. Metall. C), 2009, Vol 118, No 3, pp. 146-155; Hongyu Tian, Jian Pan, Deqing Zhu, Congcong Yang, Zhengqi Guo, Yuxiao Xue, “Improved beneficiation of nickel and iron from a low-grade saprolite laterite by addition of limonitic laterite ore and CaCO3”, J. of Materials Research and Technology, 2020, Vol 9(2), pp.2578-2589)
- It is not clear which is the advantage of FeNi-ceramic material production. The process in the introduction is referred as “green metallurgical”, however this consideration is not evidenced. High temperatures (up to 1450oC) and complex processes (reductive atmosphere, fast cooling) are required.
- Why C3S specifically should be formed instead of other phases?
- The “final product” is composed by a mixture of inter-diffused FeNi and ceramic phases as it is revealed in SEM images. Which is the meaning of the production of such a material? A post-processing step is required for the separation of FeNi by C3S. The separation will be a costly process, while the separation efficiency will be poor due to sintered structure of the material.
- The term “coupled preparation of ferronickel and cementitious material”, as it is used in the title and in the MS, is confusing. The term production of a complex FeNi/cementitious material though a controlled reduction process would be scientifically more appropriate.
Reviewer 3 Report
Review
of the Manuscript ID: materials-960546
Title: Coupled Preparation of Ferronickel and Cementitious Material from Laterite Nickel Ores.
Authors: Ruimeng Shi, Xiaoming Li, Yaru Cui, Junxue Zhao, Chong Zou, Guibao Qiu.
Authors describe results of the coupled preparation of ferronickel and cementitious materials with calcium silicate and calcium aluminate as the main phases. They formulated impressive recommendations and presented also results of the same experimental process. Derived results can present some interest for working companies. Everything is OK but I hope that authors consider their paper as a scientific article. However in an existing form, the article reminds the culinary recipe: one can trust or reject, since no explanations in it are present. At the same time, it is accepted in the Science to doubt of any statements, which have been not supported by evident proofs. Authors used FactSage software to perform thermodynamic calculations and analyze the reduction of nickel and iron and the effect of reduction on the formation of tricalcium silicate and tricalcium aluminate. Computer predicted some optimal regime. Is it valid or we have a failure in the work of the program? Even in the case of an analytical solution of any equations it is necessary to have some simple explanation of derived results on the base of physical nature of investigated processes. For the case of computer calculations it is especially necessary. I think that it would be very nice for the article if some simple estimations would be made on the ground of basic properties of iron and nickel and included into the text to confirm declared recommendations.
I recommend this article for publication in “Materials” after corresponding successful corrections and believe that this work will be useful for many readers who are interested in extraction from ore of various substances.
Round 2
Reviewer 2 Report
I received the “Coupled Preparation of Ferronickel and Cementitious Material from Laterite Nickel Ores”, Shi et al., however i notice that None of my previous comments were taken into account in the revised manuscript.
Minor reviews were performed in the “introduction” part. Below a part of the revised introduction is quoted:
“Due to the direct reduction of rotary kiln, tailings yield is large and slag cannot be directly used, which increases the cost of subsequent treatment of nickel slag [37-45]. However, in the process of fire smelting of nickel-iron from laterite, the gangue components in the raw ore are comprehensively treated and applied to high value-added materials, so as to realize the green metallurgical utilization of laterite nickel, which has the characteristics of low cost, high added value and short flowing time. In this study, while preparing nickel iron from carbon base of laterite nickel ore by direct reduction, the gelling materials with silicate tricalcium silicate (C3S) and aluminate tricalcium (C3A) as main components were prepared from coal gangue. The coupling of these two reaction processes was investigate experimentally”.
It is not documented by the authors, even on a theoretical basis, why their suggested methodology is less costly and more environmentally friendly taking into consideration that further processing steps including; crushing, magnetic separation and smelting will be necessary for the separation of FeNi by the aluminosilicate phases.
Therefore, prior to the publication of the MS, I propose a more detailed description of the work’s aim, possible advantages of the described methodology in comparison to the conventional laterite pyrometallurgy and how the mixed FeNi/aluminosilicate product will be further processed.
